# Feasibility of Canine Adenovirus Type 2 (CAV2) Based Vector for the Locus Coeruleus Optogenetic Activation in Non-Transgenic Rats: Implications for Functional Studies

**DOI:** 10.3390/brainsci12070904

**Published:** 2022-07-10

**Authors:** Anna Kabanova, Elena Cavani, Nikos K. Logothetis, Oxana Eschenko

**Affiliations:** 1Department of Physiology of Cognitive Processes, Max Planck Institute for Biological Cybernetics, 72076 Tübingen, Germany; anna.kabanova@tuebingen.mpg.de (A.K.); elena.cavani@tuebingen.mpg.de (E.C.); nikos.logothetis@tuebingen.mpg.de (N.K.L.); 2Division of Imaging Science and Biomedical Engineering, University of Manchester, Manchester M13 9PT, UK

**Keywords:** locus coeruleus, optogenetics, channel-rhodopsin, immunohistochemistry, non-transgenic rat, CAV2 virus, norepinephrine, retrograde tracing, PRSx8 promoter

## Abstract

The locus coeruleus norepinephrine (LC-NE) system modulates many visceral and cognitive functions, while LC-NE dysfunction leads to neurological and neurodegenerative conditions such as sleep disorders, depression, ADHD, or Alzheimer’s disease. Innovative viral-vector and gene-engineering technology combined with the availability of cell-specific promoters enabled regional targeting and selective control over phenotypically specific populations of neurons. We transduced the LC-NE neurons in adult male rats by delivering the canine adenovirus type 2-based vector carrying the NE-specific promoter PRSx8 and a light-sensitive channelrhodopsin-2 receptor (ChR2) directly in the LC or retrogradely from the LC targets. The highest ChR2 expression level was achieved when the virus was delivered medially to the trigeminal pathway and ~100 μm lateral to the LC. The injections close or directly in the LC compromised the tissue integrity and NE cell phenotype. Retrograde labeling was more optimal given the transduction of projection-selective subpopulations. Our results highlight a limited inference of ChR2 expression from representative cases to the entire population of targeted cells. The actual fraction of manipulated neurons appears most essential for an adequate interpretation of the study outcome. The actual fraction of manipulated neurons appears most essential for an adequate interpretation of the study outcome. Thus, besides the cell-type specificity and the transduction efficiency, the between-subject variability in the proportion of the remaining viral-transduced targeted cell population must be considered in any functional connectivity study.

## 1. Introduction

The brainstem locus coeruleus norepinephrine (LC-NE) system comprises a set of small neuronal clusters (nuclei) located in the pons and medulla [1]. Among these NE-producing brainstem nuclei, the LC presents a primary source of NE in the forebrain [2,3]. The LC-NE system regulates many visceral and cognitive functions [4,5] and has been a focus of multidisciplinary research over the last several decades [6]. Pathological changes in the LC-NE system have been attributed to various neuropsychiatric and neurodegenerative disorders such as depression, ADHD, circadian rhythm and sleep disturbances, post-traumatic stress disorders, epilepsy, Alzheimer’s, and Parkinson’s diseases [7,8,9,10,11]. 

The small size of the LC core, diffuse forebrain projections of NE neurons, and a relatively low innervation density in the cortex have always presented a technical challenge to selective manipulation of the LC-NE system, particularly in behaving animals. Modern gene technologies such as the Cre/loxP system, viral vectors containing cell-specific promoters, and genome editing have enabled the generation of transgenic and conditional knockout mouse models permitting loss or gain of function experiments to explore the role of selected neuronal populations and pathways [12,13]. Transgenic mice lacking (knockout) or carrying (knockin) a particular gene have been instrumental in a myriad of functional studies despite some obvious limitations of this experimental approach [14,15]. Recent advances in molecular engineering have allowed spatially and temporally controlled gene manipulations. Namely, spatially-controlled gene expression/knockout is achieved by the use of cell type-specific promoters to drive Cre recombinase-mediated excision of the loxP sites flanking a functionally essential genomic sequence [16]. In addition, gene expression/knockout can be regulated by exogenous ligands such as tamoxifen or tetracycline in CreERT2 or tetO-, and TRE-Cre mouse models, respectively [14,15]. However, the successful application of this technology depends on the availability of tissue-specific regulatory elements to drive Cre expression. Over the last several decades, numerous Cre-driver mouse lines have been generated and used for research in different fields, including neuroscience [17]. In contrast to mouse models, very few transgenic rat models are currently available [18] thus restricting the use of this state-of-the-art technology in the rat. Moreover, even with advanced genetic tools such as CRISPR/Cas9, producing genetically modified animal models (including non-human primates) remains costly and time-consuming [19].

The optogenetic tools present an alternative approach to the loss or gain of function experiments in non-transgenic animals. This method employs the use of exogenous light-sensitive receptors to transiently and bidirectionally modulate selective neural circuits [20]. The viral delivery of ChR2 under the control of cell-specific promoters provides cell-type specificity and has been successfully applied to a wide range of neuron populations to modulate neuronal dynamics and animal behavior [20]. The adeno-associated viruses (AAV) have been commonly used to deliver the transgenes into neuronal somata and axon terminals due to their negligible immunogenicity. However, AAVs show low tropism in non-transgenic animals [13,21,22,23]. The canine adenovirus type 2 (CAV2) has been suggested for use in neuroscience due to its capability of preferentially transducing neuronal axon terminals and somata via a retrograde transport [24,25,26]. 

Several studies have successfully used the CAV2 virus carrying a cell-type-specific PRSx8 promoter [27] for transduction of the LC-NE neurons and NE-containing fibers in non-transgenic rats [28,29,30,31,32,33,34]. Despite the great success in using both AAV- and CAV2-based viral vectors for various applications in neuroscience in transgenic and non-transgenic animals, their use requires a critical assessment of viral serotype and titer, a suitable promoter for cell-selective transduction of viral particles, as well as optimal injection parameters. In this study, we quantitatively assessed the selectivity of the NE neuron-specific PRSx8 promoter and evaluated the CAV2-based transgene delivery into LC-NE neurons using both local and retrograde transduction strategies. Our results highlight the advantages and drawbacks of the optogenetic approach employing CAV2-based viral vectors for modulation of the LC-NE system in non-transgenic rats.

## 2. Materials and Methods

### 2.1. Subjects

A total of 43 adult male Sprague Dawley rats (Charles River Laboratory, Sulzfeld, Germany) weighing 250–300 g at the beginning of the experiment were used. All experiments were conducted in full compliance with the guidelines of the EU Directive on the protection of animals used for scientific purposes (2010/63/EU). The local ethical commission (§15 TierSchG) reviewed and approved the study protocol (KY04-19G). Animals were kept under environmentally controlled conditions: 12 h light/dark cycles, temperature at 20–23 °C and humidity at 40–60%, with food and water ad libitum. All animals were group-housed, except for a 1-week post-surgery recovery when the animals were single-housed. Cages were enriched with paper nesting material, small wooden objects, and cardboard tunnels.

### 2.2. Surgery and Virus Injection

Animals were anesthetized with a mixture of medical oxygen and isoflurane vapor (5% induction, 1.5–2% maintenance) and after ensuring a lack of responses to a hind paw pinch were secured in a stereotaxic frame (David Kopf Instruments, Tujunga, CA, USA). The skull was leveled in such a way that the difference between bregma and lambda was less than 0.2 mm. Blood oxygenation and heart rate were monitored using a pulse oximeter (Nonin 8600 V, Nonin Medical, Inc., Plymouth, MN, USA); supplementary oxygen was provided to maintain the blood oxygen level above 90%. Throughout the entire anesthesia period, rats were kept on the heating pads to maintain a body temperature of ~37 °C. Under sterile conditions and additional local anesthesia (Lidocaine 2%, 5 mg/mL/kg, s.c.), a small incision was made along the midline of the skull and a small unilateral craniotomy was drilled above the target region. For transduction of LC-NE neurons, we used the CAV2 vector (PVM-IGMM, BioCampus Montpellier, France) encoding a gene cassette for ChR2 expression under the control of PRSx8, a synthetic dopamine beta-hydroxylase(DbH) promoter [27,35]. The CAV2-PRSx8-ChR2-mCherry was injected using a programmable nanoliter injector Nanoject III (Drummond Scientific Company, Broomall, PA) and a glass capillary pipette (40 μm tip). The stereotaxic coordinates for the LC were ~4.0–4.3 mm posterior to lambda and 1.35–1.5 mm mediolateral [36]. To ensure the accuracy of virus injection in the proximity to the LC core, we used electrophysiological verification of the presence of neural activity of LC-NE neurons and mesencephalic trigeminal neurons as described in detail elsewhere [37]. Following previous reports [31,38], a total of 1μL of the viral suspension containing 1 × 10^9^ viral particles (stock virus titer 1 × 10^12^ vp/mL) was injected at 3 dorsoventral coordinates (−6.2, −5.9, and −5.6 mm). All injections in the LC were unilateral (*n* = 23) and performed at a 15° angle (Figure 1a). In the forebrain, both unilateral and bilateral injections were made. We performed a total of 12 injections (*n* = 8 rats) in the anterior cingulate cortex (ACC), 9 injections (*n* = 6 rats) in the prefrontal cortex (PFC), and 8 injections (*n* = 6 rats) in the dorsal hippocampus (dHPC). At each site, the injection volume was split for 500 nl (1.5 × 10^9^ viral particles) and injected at a constant speed of 50 nL/min. The following coordinates (from Bregma) were used for the ACC (AP: 1.7–2.9 mm, ML: 0.4 mm, DV: 1.4 mm), for the PFC (AP: 3.2–4.2 mm, ML: 0.5–0.8 mm, DV: 3.5 mm), and the HPC (AP: −2.3–4.0 mm, ML: 1.9–3.4 mm, DV: 2.0 mm). Figure 2 illustrates the injection placements. After each injection, the glass capillary pipette was maintained at its injection position for 5 min to minimize the upward flow of viral solution after raising the glass pipette and then slowly (1 to 3 min) retracted. At the end of the surgery, a two-component silicone gel (Kwik-Sil: World Precision Instruments, Sarasota, FL, USA) was applied above the craniotomy for brain protection. The skin incision was sutured, and the animals received an injection of analgesic (2.5 mg/kg, s.c.; Rimadyl, Zoetis) and antibiotic (5.0 mg/kg, s.c.; Baytril^®^) before the rat’s awakening from anesthesia. The antibiotic and analgesic treatment was applied for 5 days.

### 2.3. In Vivo Electrophysiology, Optical Stimulation, and Electrophysiology Data Analysis

Electrophysiological recordings combined with optical stimulation were performed 2 to 4 weeks after the virus injection in isoflurane-anesthetized rats. Animal preparation, anesthesia induction, and maintenance were performed as described above. The LC was targeted stereotaxically by a homemade optrode consisting of an optic fiber (125 µm diameter, standard network fiber patch cable SC-UPC OM3 50/125 type) glued to the Neuronexus 16-channel array with 40 μm spacing between the electrode contacts (Neuronexus Technologies, Ann Arbor, MI, USA). The optic fiber was positioned at ~1 mm from the electrode tip. The optic fiber was connected to a 470 nm blue light diode laser (LDR-0470 Laserglow Technologies, Toronto, ON, Canada) via ferrule. The laser pulses were controlled digitally by using a QNX-based operating system (QNS, Waterloo, Ontario, Canada). The Neuronexus electrode array was connected to the acquisition system (Cambridge Electronic Design, Cambridge, UK) via in-house designed and built equipment (plug, preamplifier, and amplifier). After LC spiking activity was detected, laser pulses (10–30 ms) were applied at 1 to 20 Hz with a laser intensity of 15 to 80 mW. The broad-band (0.1 Hz–5 kHz) extracellular signals were amplified (10^10^ V/A) and digitized at 24 kHz using CED Power1401 mkII converter and Spike2 data acquisition software (Cambridge Electronic Design, Cambridge, UK). To additionally verify the electrode placement in the LC, somatosensory stimulation consisting of a brief train (100 ms at 100 Hz) of biphasic electric pulses (2 mA) was delivered via two stainless steel needles inserted subcutaneously (~1 cm apart) in the front paw contralateral to the LC recording site. The electrical current was delivered using an in-house designed and built biphasic stimulus isolator. The stimulation parameters were digitally controlled by a QNX (QNS, Waterloo, Ontario, Canada) operating system and transformed by a digital-to-analog converter (NI 6713 DAQ Card, National Instruments, Austin, TX, USA) driving the isolator. The neurochemical nature of the recorded units has be was additionally verified by silencing spiking activity by clonidine (0.05 mg/kg, i.p.) injection [37].

The broad-band (0.1 Hz–8 kHz) extracellular signal was high-pass (600 Hz) filtered and multiunit spiking activity extracted. When the recording quality permitted, a single unit activity was isolated using a template matching using Spike2 data analysis software (Cambridge Electronic Design, Cambridge, UK). Peri-stimulus time histograms (PSTHs) were computed around the stimulation onset.

### 2.4. Immunohistochemistry, Imaging, and Cell Quantification

The immunohistochemical analysis was performed 1 to 4 weeks after the virus injection. At the end of the virus transduction period, animals were euthanized with a lethal dose of sodium pentobarbital (100 mg/kg i.p.; Narcoren^®^) and perfused transcardially, first with 0.9% saline, followed by 4% paraformaldehyde in 0.1 M phosphate buffer (PBS, pH 7.4). Brains were removed and post-fixed in the same fixative overnight, then transferred to 30% sucrose solution in 1xPBS at 4 °C until the tissue sank. Brains were serially sectioned at the coronal plane with a sliding microtome (Microm HM 440E, Walldorf, Germany) at 40 µm thickness. Sections were immersed in 1xPBS and stored at 4 °C until further use.

To confirm the virus transduction in the LC-NE neurons and their axons, double-labeled immunostaining for DbH (a marker for a NE-containing neuron) and mCherry (a marker for ChR2) was performed. The brain sections containing the brain regions of interest were selected and kept for 1 h at room temperature (RT) in 10% normal donkey serum (Jackson ImmunoResearch) and 0.2% Triton X-100 prepared in 1xPBS (PBST) and then incubated overnight at 4 °C with primary antibodies (mouse DbH-specific antibody, 1:1000, Merck, Darmstadt, Germany), diluted in blocking buffer (3% NDS/0.2% PBST). After additional washing with 0.2% PBST, the sections were incubated with appropriate donkey anti-mouse/rabbit secondary antibodies conjugated to Alexa Fluor 488 (1:500, Invitrogen) and Cy3 (1:200, Jackson Immunoresearch West Grove, PA, USA) for 2 h at RT, followed by extensive washing with 0.2% PBST. To label the neuron nuclei, all sections were counterstained with Hoechst 33,258 (1:10,000, Sigma-Aldrich, Burlington, MA, USA) for 10 min, followed by 3 additional rinses with 0.2% PBST. The brain sections were mounted on microscope slides, covered with Aqua-Polymount mounting medium (Polysciences, Inc., Warrington, PA, USA), and imaged on a Zeiss Axio observer using a 20× objective and Z-Stack setup (Axiovision, Zeiss, Oberkochen, Germany). To verify the presence of virus-containing LC-NE, Z-stack images of the selected forebrain areas were acquired on a Zeiss Axio observer using a 40× objective and ApoTome Setup and the sections were consequently visually examined for the presence of double-labeled (DbH- and mCherry-positive) fibers. 

Maximum-intensity projections were imported into ImageJ software for cell counting. The cells were counted bilaterally in injected LC and non-injected LC and at each rostro-caudal level (from Bregma: 10.1 mm, 9.8 mm, 9.5 mm). First, all distinguishable NE neurons (DbH-positive) with a visible nucleus (labeled with Hoechst) were counted. Then the DbH-positive and Hoechst-marked cells were transferred to a merged image of the green and red channels, so the double-labeled cells for both DbH and mCherry could be visualized (Figure 2b,c). The number of double-labeled cells was extracted from all three rostrocaudal sections to estimate the total number of virus-transduced LC-NE neurons and their spatial distribution. To quantify the ChR2 expression level in the LC-NE neurons, the number of double-labeled (DbH- and mCherry-positive) cells was normalized to the total number of DbH-positive cells. To validate the cell specificity of the PRSx8 promoter the number of double-labeled (DbH- and mCherry-positive) neurons was normalized to the total number of the mCherry-positive cells. To characterize potential tissue damage due to virus neurotoxicity and the electrode placement, the cell count in the virus-transduced LC was compared to the contralateral intact LC. In the case of retrograde labeling, the number of LC-NE cells (DbH-positive) was compared to the average number of NE cells in intact LC estimated from the quantification of 30 brain samples. Finally, to determine the population size of LC-NE neurons that could potentially be manipulated in the functional experiment (e.g., transduced and remaining LC-NE cells), the number of double-labeled (DbH- and mCherry-positive) neurons was normalized to the number of DbH-positive cells in the intact LC. In the case of retrograde LC transduction, the number of double-labeled neurons was normalized to the average number of DbH-positive cells in the corresponding section. 

## 3. Results

### 3.1. Transduction Efficiency of the CAV2-PRSx8-ChR2-mCherry for LC-NE Neurons

To selectively transduce LC-NE neurons, we used the CAV2-based viral construct carrying ChR2 fused with mCherry fluorescent indicator, whose expression is under the control of the DbH promoter PRSx8 [27]. We targeted the virus injection close to the LC cell bodies. Briefly, the stereotaxic coordinates for the virus injection were adjusted for each rat based on the electrophysiology-guided mapping of the LC neurons and mesencephalic trigeminal tract (see Methods for details). For all injections, we used the same virus concentration and injection volume of 1 μL, which was split into three injection sites. The transduction time varied from 1 to 4 weeks. 

After the virus transduction period, we confirmed the functionality of the virus construct for the light-induced activation of the LC-NE neurons using in vivo electrophysiology combined with optical LC stimulation in seven rats (see Methods for details). Figure 1c–e show a discharge of LC-NE neurons in response to laser light. Depending on the exact placement of the optical fiber, the laser stimulation (10–30 ms pulses) with an intensity of 15 to 80 mW reliably elicited the LC spikes with a latency of ~10 ms. Importantly, a biphasic LC discharge profile with excitation followed by inhibition resembled a naturalistic response of LC-NE neurons to salient stimuli [39]. The neurochemical nature of the recorded neurons was additionally verified by the cessation of firing after clonidine [37]. As described in detail below, post hoc histological analysis confirmed that a failure of optical stimulation in other cases was related to mistargeting the LC, a low number of labeled LC neurons, or tissue damage within the LC area. 

Using the immuno-stained histological material (see Methods), we assessed the ChR2 expression, promoter specificity, and tissue integrity. For histological examination, we selected LC-containing sections with no or minimal tissue damage. Since the virus injection was performed unilaterally, the intact (non-injected) contralateral LC was used as a reference for the quantitative analysis. Promoter specificity was evaluated as a proportion of LC-NE neurons co-labeled with mCherry to mCherry-positive neurons, whereas transduction efficiency was calculated as the percentage of mCherry-positive neurons to DbH-positive ones. 

Out of 23 cases, three cases were excluded from quantitative analysis. Specifically, in one rat (case 39.4), there was severe tissue damage in the LC area. Besides, none of the DbH-positive neurons could be detected in the vicinity of the virus injection site. Although an attempt at electrophysiological recording and optical stimulation has been made in this rat, the post-hoc histological examination revealed that the electrode passed ~0.1 mm medial to the LC core; therefore, any mechanical tissue damage within the LC due to the electrode penetration could be excluded. Although we did not label the virus injection site in this case, based on the LC mapping results, we assumed that the virus was injected directly into the LC core. Thus, a combination of the glass pipette incision directly to the LC core and a relatively high virus concentration within the injection site likely resulted in severe neuronal toxicity and triggered cell apoptosis and tissue lesion. Another case (59.2) had only ~4% double-labeled (DbH and mCherry positive) neurons, while the DbH staining was present within the entire LC nucleus. The post-hoc histological examination suggested that in this case, inefficient virus transduction was likely due to the injection site being too lateral (>100 µm) from the LC core. One more case (116.2) was excluded from the analysis because of unsuccessful immunostaining.

In the remaining 20 cases, a preliminary examination of the histological material revealed substantial inter-subject variability in the number of virus-transduced LC cells (Table 1). Although we used the same viral vector and the standard injection procedure, the proportion of labeled LC neurons varied from 39.06% to 98.9% and it did not appear to depend on the virus transduction time (independent samples Kruskal–Wallis test, ns; Figure 2f). A bimodal distribution of the number of double-labeled LC neurons separated cases with high (>70%, *n* = 12) and intermediate (39% to 61%, *n* = 8) ChR2 expression level (Figure 2g). Figure 2b,c show representative cases with a high and intermediate transduction rate. On average, 85.6 ± 2.8% and 48.9 ± 2.4% of all DbH-positive cells in the LC were also mCherry-positive in the subgroups with high and intermediate labeling, respectively. The efficient virus transduction was also confirmed by the virus expression in the LC axonal terminals. The mCherry-positive axonal fibers were detected in the LC distal targets, such as the PFC (Figure 2e–e”), dHPC, and basolateral amygdala already one week after virus injection (data not shown). Based on a systematic microscope examination of histological material it was evident that the virus expression in the LC axonal terminals positively correlated with the LC transduction rate and/or integrity. However, because of the intermingled distribution of NE projecting neuron populations within the LC, it seemed impossible to determine a mass of LC axonal terminal loss in one or another LC target region.

Despite substantial between-subject variability in the proportion of virus transduced LC-NE neurons compared to the total number of the DbH-positive cells in the intact LC, the expression of the synthetic PRSx8 promoter, was highly selective and largely restricted to the NE neurons (87.4 ± 4.7% of all mCherry-positive cells within LC). More importantly, we systematically observed downregulation of DbH expression in the LC-NE neurons, whereas mCherry expression was present (Figure 2d’–d”’). The DbH downregulation could be a result of competitive interactions of synthetic PRSx8 promoter and endogenous Phox2 genes [40]; both factors likely lead to a loss of NE cell phenotype. A decreased level of DbH expression may indicate a loss of the NE phenotype of LC neurons [40]. The DbH expression downregulation appeared to positively correlate with the transduction rate as it depended on the virus injection site. As shown in Table 1, the proximity of the injection site to the LC was associated with the DbH downregulation. There was no apparent dependency on the transduction time (Table 1). 

Besides the inter-subject difference in the number of labeled LC neurons, the virus injection differentially affected the LC integrity. The proportion of preserved LC-NE cells (compared to the intact contralateral hemisphere) showed a bimodal distribution with a split at around ~50% (Figure 2h,i). In the subgroup with the high number of double-labeled LC neurons, the LC was well preserved in 7 out of 12 rats (58.3%). In the other five cases, the LC integrity was substantially compromised as only 7 to 30% of neurons within the LC area were DbH-positive. In contrast, almost in all cases (87.5%) with an intermediate labeling, most of the LC nucleus was well preserved (77.2 ± 9.6% of LC-NE neurons). The LC cell loss could be related to the toxicity of a relatively high virus titer used in the present study or to the mechanical tissue damage due to the LC targeting with the electrode or injection pipette. Retrospective analysis showed that in four cases there was partial tissue damage due to multiple penetrations of the electrode in the LC area during the LC mapping and/or functional validation of the ChR2-expression on LC-NE neurons. In the other 14 cases with compromised LC integrity, an NE neuron loss could be attributed to viral neurotoxicity, as no mechanical damage to the brain tissue was present. 

Thus, there appears to be an interplay between the transduction efficiency, the LC integrity, and the maintenance of the NE neuron phenotype. The reconstruction of the virus injection sites indicated that the injections in the LC proximity, but ~100–150 μm outside the LC border and lateral to the mesencephalic trigeminal nucleus (Me5), typically resulted in a good outcome (Figure 1a and Figure 2c; Table 1). The injections placed within 50 μm from the LC cell bodies or directly in the LC core resulted in substantial NE neuron loss and/or DbH downregulation (Figure 1a and Figure 2b,d; Table 1). Notably, LC-NE cell death was evident as early as one week after injection (see case 112.2, Table 1).

### 3.2. Retrograde Transduction of the Projection-Specific LC-NE Neurons with CAV2-PRSx8-ChR2-mCherry

To retrogradely label LC-NE neurons, we injected the CAV2-PRSx8 virus construct encoding ChR2-mCherry into one of three major forebrain targets of the LC-NE neurons (Figure 3a, upper panel). Typically, multiple injections of 1.5μL total volume and 1.5 × 10^9^ pP were performed unilaterally or bilaterally along the anteroposterior axis of the region of interest (Figure 3a, lower panel; see Methods for more details). In total, we performed twelve injections in eight rats in the ACC, nine injections (*n* = six rats) in the PFC, and eight injections (*n* = six rats) in the dHPC. Notably, mCherry-labeling of fibers and cell bodies along the glass capillary path was clearly seen. Figure 3b”’–d”’ illustrates the sparse labeling of neurons in the ACC, PFC, and dHPC within a 50 μm distance from the injection cannula track. Such labeling could indicate the virus capacity for the non-specific labeling of non-NE neurons. 

The number of retrogradely labeled LC neurons was estimated at 2 to 8 weeks after the virus injection (Figure 3b–d). The number of double-labeled (DbH- and mCherry-positive) LC neurons was normalized to all DbH-positive neurons in the intact LC. The spatial pattern of double-labeled LC-NA neurons projecting to the ACC and PFC was sparse without any particular neuron clustering (Figure 3b,c). The ‘salt-and-pepper’ pattern for the mPFC-projecting LC-NE neurons has been previously described in rats using the horseradish peroxidase, a ‘classical’ retrograde tracer [41] and reproduced more recently in mice using viral-based tracing [42,43]. The hippocampal-projecting LC-NE neurons were mostly scattered in the dorsal part of the nucleus, which is consistent with previous observations [44]. 

The ChR2 expression was preserved in LC-NE neurons for up to 8 weeks post-injection, making this viral vector suitable for longitudinal studies once the injection procedures are optimized. Most importantly, visual examination of the histology sections showed no signs of cell death, tissue damage, or DbH downregulation within the borders of the LC nucleus, which was, in contrast, the case after the virus injections into the LC area.

Out of a total of 29 injections, two cases were excluded due to technical reasons as no reliable injection volume could be delivered or the quality of immunohistochemical staining was not sufficient for analysis. Table 2 provides an overview of the entire dataset included for the quantitative histological analysis. There was substantial variability in the proportion of the retrogradely-labeled LC neurons (Figure 3e). Consistent with the predominant ipsilateral connectivity of the LC with the cortical regions, including the dHPC [3], the proportion of virus-labeled LC neurons was substantially lower in the LC nucleus contralateral to the injection site and varied from none to 3.19% (Figure 3e, right panel). At the same time, cases with low ipsilateral labeling were also present (Figure 3e, left). Close inspection of the injection sites revealed that in all cases with low ipsilateral labeling (<3.5% neurons), the injection volume was insufficient to cover the entire target region. Several methodological aspects could have contributed to the delivery of a lower volume. For example, extensive intra-brain bleeding could have clogged the pipette tip and led to a lower volume injected. In some cases, the skull midline was not a reliable landmark for the hemispheres’ border; therefore, in such cases, the injections were misaligned. Taking the above-mentioned factors into account, the cases with low ipsilateral labeling were excluded from further quantitative analysis using an arbitrary threshold of 3.5% (Figure 3e, horizontal dotted line). 

We next quantified the labeling in the LC nucleus ipsilateral to the injection site. Despite a tendency for a larger population of the prefrontal-projecting compared to the hippocampal-projecting LC-NE neurons (Figure 3f), an overall number of retrogradely labeled LC-NE neurons was comparable across three brain regions (one-way ANOVA, F_1,20_ = 1.8, *p* = 0.19). Furthermore, the virus transduction of LC neurons did not increase significantly with time (F_1,20_ = 0.68, *p* = 0.64). Moreover, cases with extremely low retrograde labeling were observed even after a prolonged transduction time, regardless of the injection site. The most efficient transduction (>10 %) was achieved exclusively after 6 to 8 weeks and only for the prefrontal-projecting LC neurons (Figure 3g). The retrograde labeling from the dHPC resulted in a systematically lower number of transduced LC-NE neurons (<10% of the LC-NE population). Somewhat low retrograde LC neuron labeling from the dHPC was rather unexpected, given a high density of NE-containing fibers originating from the LC [33,45]. At the same time, a relatively small proportion (~5%) of retrogradely labeled hippocampal-projecting LC neurons has been previously reported [46]. The whole-cell reconstruction of the LC neurons projecting to different cortical targets may clarify this discrepancy in the future. Overall, our results are consistent with other studies reporting a ‘salt-and-pepper’ spatial distribution of the prefrontal-projecting LC neurons [42,43] but see [47], and predominant dorsal distribution of the hippocampal-projecting LC neurons [41]. 

Thus, the retrograde transport of the CAV2-PRSx8-ChR2-mCherry from the distal forebrain targets of the LC resulted in selective labeling of projection-specific LC-NE neurons. Quantitative analysis revealed a larger population of the ACC-projecting LC-NE neurons. The LC neuron transduction was the highest after 6 to 8 weeks. Relatively high variability of the LC neuron labeling could be attributed to several factors including the virus titer, injection parameters (stereotaxic coordinates, injection volume), and/or variable presence of surface coxsackievirus and adenovirus receptor (CAR) on subsets of neurons (viral tropism).

## 4. Discussion

In the present study, we evaluated two strategies for the virus delivery to the LC-NE neurons in non-transgenic rats by performing quantitative histological analysis. To achieve the cell-type specificity, we delivered the CAV2-PRSx8-ChR2-mCherry viral vector, carrying the DbH promoter PRSx8, either in the LC region or in the LC forebrain targets. In the case of virus injection directly in the LC, the most efficient transduction of the LC-NE neurons was achieved when the injection was placed ~100 μm lateral to the LC core and medial to the mesencephalic trigeminal pathway (Figure 1a and Figure 2b). The optimal virus delivery resulted in the efficient transduction of almost the entire population of the LC-NE neurons already after 1 week without any detectable signs of neurotoxicity up to 4 weeks following the virus exposure. In contrast, suboptimal injections resulted in highly inconsistent ChR2 expression in LC neurons. The virus delivery within the LC core or close to the LC cell bodies caused various undesirable outcomes such as cell death, DbH downregulation, and/or tissue lesion. The virus transduction of the LC-NE neurons via retrograde transport from the ACC, PFC, or dHPC preserved the LC integrity while requiring at least 6 weeks of transduction time to achieve the maximal number of transduced neurons. Significant inter-subject variability in the number of virus-transduced LC-NE neurons inevitably affects the strength of the optogenetic activation of the affected network, which is a key factor for any functional study.

Optogenetics rapidly has become a standard tool in neuroscience research as it permits cell-type-specific targeting and temporally precise activity modulation [20]; both features are desirable for a gain or loss of function experiments. In the past decade, the optogenetic approach has been increasingly applied in biomedical research and cognitive neuroscience [48,49,50,51]. The most commonly used methods to deliver functional opsin genes are viruses, in vivo electroporation, and the generation of transgenic animals. A Cre/lox genetic approach allows for restricting viral-mediated selective expression of optogenetic genes in neurons that express Cre-recombinase. The broad use of optogenetics promoted the generation of many transgenic mouse lines, which offer important advantages such as stable and heritable transgene expression patterns. However, the generation of Cre- or Flp-dependent transgenic lines in other species including rats appears more challenging compared to mice [52,53]. Nevertheless, the optogenetic manipulation of neural circuits in non-transgenic experimental models, including rats, presents an interest for many potential users. 

Despite the predominant use of AAV-based viral vectors, the use of CAV2-based vectors is gaining its niche in neuroscience [24]. The large physical size of CAV2 (∼90 nm in diameter vs. ~22 nm of AAV) provides some advantages for targeting small brain nuclei, like the LC, as the virus particles remain near the injection site. For instance, 0.25–0.5 µL of CAV2 spread on average by 200 µm from the center of the injection site, while 0.25 µL of rAAV2-retro spread four times more [3,54]. More restricted diffusion of CAV2 has been proved useful when studying small brain structures [55]. Whereas AAVs transduce neuron soma to express gene transcripts, CARs are predominantly located on the axon surface [29] making CAV2-based vectors more suitable for the retrograde transduction of selected neuron populations both in the forebrain and spinal cord [28,30,56]. At the same time, it has been shown that CAV2 transduction is limited to some types of neurons or pathways [55], possibly due to the varying expression level of CARs in different neuron populations [57,58]. 

In the present study, we validated the efficiency of the CAV2-based vector for transduction of LC-NE neurons using direct and retrograde delivery methods in adult non-transgenic rats. For a more accurate virus delivery within the LC region, we performed electrophysiological verification of the presence of neural activity of LC-NE neurons and/or mesencephalic trigeminal neurons. Virus injections medial to the LC core were avoided due to potential virus diffusion into the 4th ventricle and dorsal to the LC due to the expectedly low transduction rate of the lateral and ventral parts of the LC. The LC mapping procedure provided high success in the LC targeting, yet resulted in relatively high variability of the proportion of transduced LC neurons. Below, we discuss in more detail some methodological pitfalls related to this transduction method.

In recent years, the optogenetic approach has been actively used to investigate with more precision the impact of NE-mediated neuromodulation of various cortical and subcortical circuits in both transgenic and non-transgenic rodent models [59]. The general enthusiasm for the obvious advantages of the optogenetic approach may have overshadowed some methodological shortcomings, which, however, may affect the interpretation of the results. One reason is an inaccurate estimate of the population size of manipulated neurons. Compared to pharmacological modulation using local drug injections, the optogenetic approach, while being more selective, in any model system cannot guarantee to affect the entire population of targeted neurons. Besides, virus-based delivery of exogenous transcripts does not lead to 100% cell penetrance even with very effective vectors, and “genetic access” of virus transcripts often depends on the presence or absence of receptors, extracellular defenses, and permissivity of the cell [60]. Among the ever-increasing number of studies employing the optogenetic tools, only a minority, if any, perform a systematic histologic evaluation of the virus delivery outcomes on the brain tissue and neuronal integrity. Most typically, the evaluation of the viral transduction is limited by presenting a representative case, and, rarely, to our best knowledge ever, the functional modulation is discussed in terms of the correlation with the tissue integrity within the modulated brain region. Similarly, the studies with optogenetic manipulation of the LC, while systematically validating the selectivity of PRSx8 promoter to drive the expression in NE neurons, reported mainly the percentage of virus transduced cells co-labeled with DbH [28,38,61]. In our study, we reproduced the previously shown high specificity of the synthetic DbH promoter PRSx8 by its expression restricted to DbH-positive neurons within the LC region (~90% colocalization). The most important result of this study was the demonstration that the high cell specificity of viral transduction is not sufficient to evaluate the cell population modulated by an exogenous ligand. Our results showed that the number of surviving opsin-expressing neurons can vary significantly between subjects, as well as from the population size in the intact brain. Thus, the viral vector specificity, the proportion of manipulated neurons, and the integrity of the targeted region shall be considered the key variables, particularly in the context of interpreting the results of functional studies. Our results suggest that detailed quantitative histological analysis is imperative for any functional study. To adequately interpret the results, the outcome of optogenetic manipulation shall be correlated with the actual number of affected neurons. 

Several different approaches have been reported for validation and evaluation of the virus transduction. One of them is measuring the fluorescent intensity of two fluorescent signals [31,61,62]. In our view, this quantification method has some disadvantages. For instance, selection of the threshold is critical to exclude all artifacts caused by fluorescent signals that can be misread as a true signal. In addition, the expression level of the fluorescent reporter (e.g., mCherry) in the neuronal cell bodies and axons often overpower the expression levels of the neuronal marker (e.g., DbH for NE). Notably, we found that in 70% of all analyzed animals, the DbH staining was overshadowed by ChR2-mCherry. Interestingly, one of the explanations for this phenomenon was that high levels of the PRSx8 promoter may disturb normal LC-NE physiology due to the sequestration of Phox2 proteins at the Phox2 binding sites in the promoter region [62]. Other colleagues reported DbH occlusion in LC-NE neurons using Cre-dependent viral constructs driven by constitutive promoters [28,63,64,65]. Stevens and colleagues [21,62] have recently attempted to develop the optimal protocol (virus titer and volume) for the use of the CAV2 vector in the LC. In our study we confirmed the transduction efficiency of CAV2 using 1 × 10^9^ pp/μL, however, we observed downregulation or even loss of DbH expression in LC-NE neurons. Whether downregulation or loss of DbH in the LC-NE neurons resulted from high virus titer and whether it affects the LC-NE phenotype need to be further investigated in detail. Possible virus-induced alternation of the LC-NE neuron phenotype, including apoptosis or DbH expression, shall be addressed in the future studies. 

Finally, preferential expression of CARs at the axonal terminals makes the CAV2 virus more suitable for a retrograde tracing, although one limited by tropism [66,67]. In our experiments, the retrograde strategy was the most optimal as it preserved the LC integrity even after a long (up to 8 weeks) transduction time. Our results for retrograde labeling of the LC-NE neurons are consistent with the AAV-based LC projection mapping from the infralimbic cortex and amygdala [42] and the hippocampus using the horseradish peroxidase tracer [46]. Yet, in comparison with other studies employing different retrograde tracers [3,41,68], the CAV2-labeled projection-selective LC populations appear overall smaller. Besides, we observed a sparse CAV2-labeling of the cell bodies around the injection site. To our knowledge, the CAV2 virus is non-trans-synaptic [24], meaning that the use of a high titer could switch the direction of virus travel [69]. Thus, sparse labeling of non-NE cells in the cortex may indicate a somewhat limited promoter specificity due to the Phox2 binding sites outside the LC area. Remarkably, no non-specific labeling with the use of a 100-fold higher titer of CAV2 virus was reported in the spinal cord [28]. 

## 5. Conclusions

The vertebrate LC-NE system comprises a small population of NE-containing neurons with widespread projections throughout the brain. The conventional methods for manipulating neural activity, besides having serious limitations related to their spatial and temporal specificity, were particularly challenging to apply in the LC due to its small size and widespread projections. The use of cutting-edge genetic and molecular biology-based methods is currently prevalent in transgenic mouse lines and has limited application in non-transgenic animals. The results of the present study highlight the advantages and drawbacks of the optogenetic approach to upregulation of the LC-NE system in the wild-type rat using CAV2-based viral vectors with a DbH-specific promoter. Our results support the notion that the virus delivery via retrograde transport should be considered the preferred method, albeit targeting only projection-specific subpopulation of LC-NE neurons. The CAV2 virus delivery within the LC region shall be carried out with extreme caution and requires careful histological examination of the tissue integrity within the LC area. Furthermore, due to substantial between-subject variability, examination of representative samples appears insufficient for the generalization of the virus efficiency to the entire population. In our view, a quantitative histological analysis of the viral transduction of the targeted cell population in each subject included in a study is essential for an adequate interpretation of the results. Considering the actual proportion of optogenetically manipulated cells appears critical for inferring causal relationships between the manipulated neural network and the function in question. In our opinion, there is an emergent need to revise some effects of optogenetic stimulation, considering the modulation strength of the targeted neural network in terms of the number of affected neuron somas and terminal fields. Such retrospective analysis may resolve ambiguities in the interpretation of study outcomes. Thus, taking advantage of the use of optogenetic tools, one should be aware of caveats when interpreting experimental results to avoid misleading conclusions.

## Figures and Tables

**Figure 1 brainsci-12-00904-f001:**
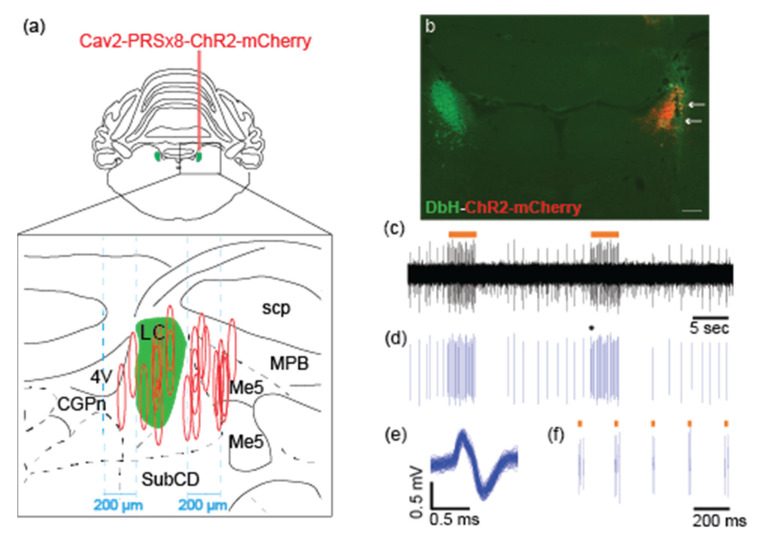
Histological and functional characterization of the CAV2-PRSx8-ChR2-mCherry virus transduction in the LC-NE neurons: (**a**) The LC targeting with the viral vector. Top, the coronal section from the rat brain atlas [36] containing the LC nucleus (Bregma −9.84 m). Bottom, reconstruction of the injection sites (*n* = 20). The LC core is highlighted in green. For each case, three injections were made in the dorsal-ventral plane with a total volume of 333 nl. Oval shapes represent an estimation of the virus spread; (**b**) A representative immunostained histological section corresponding to the highlighted region on (a, upper panel). The DbH staining (green) labels the LC-NE neurons. A specific expression of ChR2-mCherry (red) in LC-NE neurons is seen in the right hemisphere after 2 weeks after injection. Arrows indicate the electrode and optic fiber track. Scale bar: 200µm; (**c**,**d**) A representative segment of extracellular recording (**c**) and spike train of the LC single-unit showing spontaneous discharge and response evoked by optical stimulation (horizontal orange bar). Trains of laser pulses (30 ms, 80 mW at 5 Hz) were delivered above the LC dorsal border at a distance of ~1 mm from the recording site in the LC core. Note, stimulation-induced phasic LC activation was followed by autoinhibition; this pattern resembled the LC phasic response to a salient stimulus; (**e**) Spike waveform overlay of an isolated single unit shown on (**d**); (**f**) Expanded recording segment that is shown on (**d**). Note, that the LC single-unit discharge faithfully followed optical stimulation at 5 Hz, while spontaneous discharge was much slower; each laser pulse elicited 2 to 3 spikes.

**Figure 2 brainsci-12-00904-f002:**
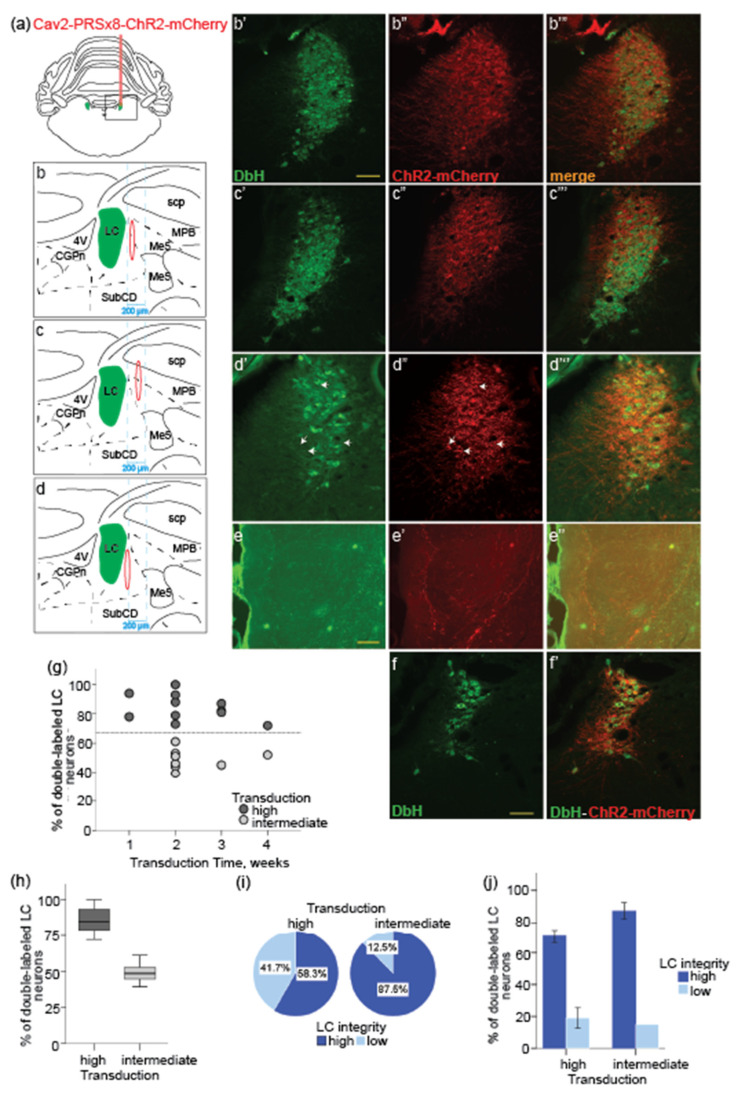
Histological analysis of the efficiency of the CAV2-PRSx8-ChR2-mCherry viral vector for transduction of LC-NE neurons and their forebrain terminal fields: Reconstruction of the injection sites (**a**–**d**) and the coronal sections at the LC plane stained with DbH (**b’**–**d’**, green), mCherry (**b”**–**d”**, red), and an overlay (**b”’**–**d”’**, yellow). Representative examples with the high (**b’**–**b”’**) and intermediate (**c’**–**c”’**) virus transduction in LC; (**d**) representative example with downregulation of DbH expression in LC neurons. Oval shapes show estimated virus spread; (**e**) DbH-stained LC axonal terminals in the PFC. (**f**,**f’**) representative example of direct virus injection into the LC core with severe cell loss. Scale bars (**b”’**–**d”’**,**e**–**e”**,**f**–**f’**): 100 µm; ((**g**) proportion of double-labeled LC neurons for each case (*n* = 20) is plotted as a function of the virus transduction time. Regardless of the transduction time, there were cases with high (>70%) and intermediate (~50%) labeling; (**h**) The average proportion of double-labeled LC neurons is shown for subgroups with the high and intermediate transduction rate; (**i**) Interdependence of the transduction efficiency and the LC integrity; (**j**) The cases with well-preserved LC (high LC integrity) had the highest number of double-labeled LC-NE neurons regardless of the transduction rate.

**Figure 3 brainsci-12-00904-f003:**
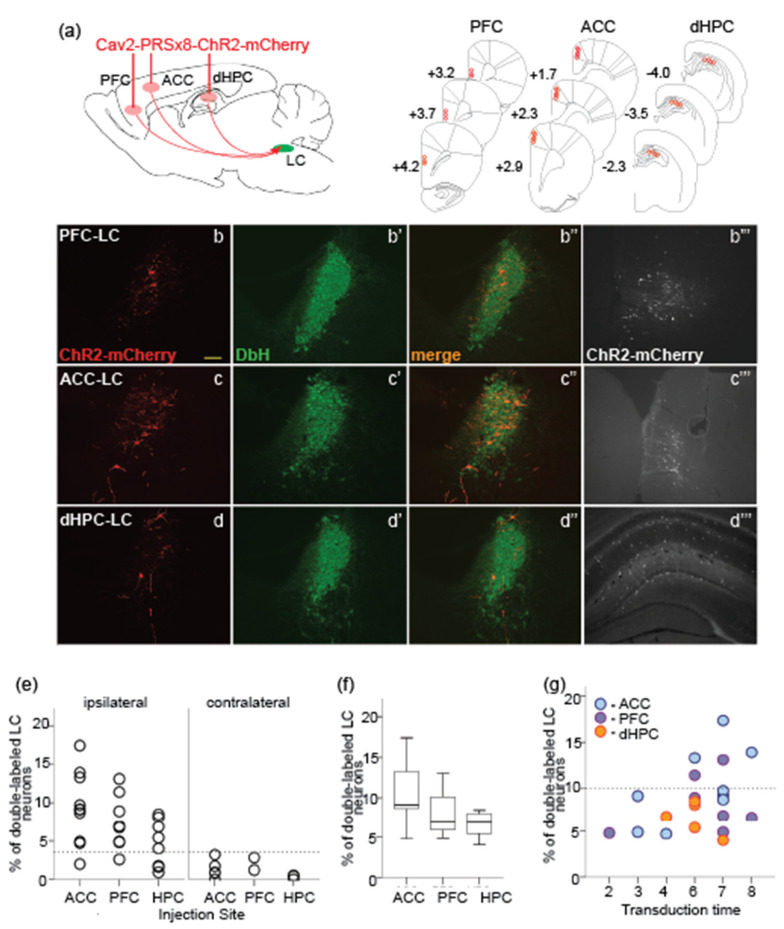
The virus transduction of the LC-NE neurons via retrograde transport from the forebrain: (**a**) Schematic of retrograde injection of the CAV2-PRSx8-ChR2-mCherry virus in three forebrain regions: the prefrontal cortex (PFC), anterior cingulate cortex (ACC) and dorsal hippocampus (dHPC); (**b**–**d**) The coronal sections at the LC plane immunostained with mCherry (**b**–**d**, red), DbH (**b’**–**d’**, green), and an overlay (**b”**–**d”**, yellow); virus-transduced neurons projecting to the PFC (**b**–**b”**), the ACC (**c**–**c”**) and the dHPC (**d**–**d”**) are labeled in red; (**b”’**–**d”’**) Coronal sections at the plane of the injection site in the PFC, ACC ad dHPC immunolabeled with mCherry (white). Note immunostained cell bodies around the virus injection site. Scale bar: 100 µm. (**e**) The distribution of transduction rates in the ipsilateral and contralateral LC; (**f**) Average transduction rate from different forebrain regions; (**g**) The transduction rate is shown after 2 to 8 weeks after the virus injection. Note the highest proportion of the virus-transduced LC neurons after 6 weeks for prefrontal, but not hippocampal injections.

**Table 1 brainsci-12-00904-t001:** Summary of histological examination of the injection sites in the LC area and quantitative analysis of immunohistological labeling.

Rat ID	TR * Time, Weeks	TR Rate, %	LC Integrity, %	Injection Site	Distance from LC	DbH Down-Regulation	Neuronal Toxicity
21.2	2	79.09	86.96	mLC	0 µm	vLC	vLC
41.2	2	88.18	82.52	Me5	<150 µm	dLC	dLC
102.2	1	78.15	73.91	Me5	≤150 µm	dLC	pLC
75.2	3	86.55	72.15	Me5	<150 µm	v-lLC	vLC
64.1	4	72.32	63.28	Me5	<150 µm	dLC	vLC
102.1	3	81.53	61.81	LC/Me5	<100 µm	LC, mosaic	LC. mosaic
36.2	2	92.97	54.01	LC/Me5	<100 µm	vLC, lLC	vLC, pLC
26.2	2	100	37.79	vLC	0 µm	mLC	vLC
112.2	1	93.75	29.09	vLC	0 µm	dLC	vLC
127.2	2	100	14.49	cLC	0 µm	dLC	dLC, vLC
96.1	2	73.33	6.88	cLC	0 µm	LC	LC
71.1	3	81.25	7.21	cLC	0 µm	none	cLC
33.2	2	52.6	98.5	Me5	≥150 µm	none	none
18.1	4	51.7	98.9	vLC	≥150 µm	none	none
13.1	2	43.77	97.23	Me5	≥150 µm	none	none
9.2	2	42.19	81.7	Me5	≥150 µm	none	LC, mosaic
75.1	3	44.63	87.19	Me5	≥150 µm	none	LC, mosaic
127.1	2	60.71	68.85	CGPn	≤100 µm	none	aLC, vLC
1103	2	39.06	67.72	dLC	0 µm	dLC	dLC
39.3	2	51.35	15.29	cLC	0 µm	dLC, mosaic	pLC
59.2	3	4.26	98.1	MPB/Me5	>150 µm	none	none
39.4	3	n/a	n/a	cLC	0 µm	n/a	n/a
116.2	3	n/a	n/a	n/a	n/a	n/a	n/a

* TR = transduction; TR rate is calculated as percent of double-labeled neurons (DbH- and mCherry-positive), LC—locus coeruleus nucleus, cLC—LC core, aLC—anterior LC, mLC—medial LC, lLC—lateral LC, dLC—dorsal LC, vLC—ventral LC, pLC—posterior LC; v-m: injections ventromedial to LC, CGPn: central gray of the pons, Me5: mesencephalic trigeminal nucleus, MPB: medial parabrachial nucleus.

**Table 2 brainsci-12-00904-t002:** Summary of the LC-NE neuron transduction via retrograde transport from the forebrain.

Brain Region	Rat ID	Injection Hemisphere	TR Time, Weeks	Labeling Hemisphere	Infected LC Neurons, %
Anterior Cingulate Cortex, ACC	2	right	4	ipsilateral	4.70
	2	left	4	ipsilateral	n/a
	18.2	right	6	ipsilateral	13.30
	18.2	left	6	ipsilateral	1.96 *
	116.1	right	3	ipsilateral	4.90
	116.1	left	3	ipsilateral	9.10
	21.1	right	8	ipsilateral	13.90
	21.1	n/a	8	contralateral	0.83
	39.1	right	7	ipsilateral	17.40
	39.1	n/a	7	contralateral	1.70
	39.2	right	7	ipsilateral	9.17
	39.2	n/a	7	contralateral	3.19
	54.1	right	7	ipsilateral	9.70
	54.1	n/a	7	contralateral	0.90
	106.2	right	7	ipsilateral	8.70
	106.2	n/a	7	contralateral	0.00
Prefrontal Cortex, PFC	9.1	right	2	ipsilateral	2.56 *
	9.1	left	2	ipsilateral	4.80
	20.2	right	6	ipsilateral	8.90
	20.2	left	6	ipsilateral	n/a
	54.3	right	7	ipsilateral	4.90
	54.3	left	7	ipsilateral	6.90
	43.1	right	7	ipsilateral	13.10
	43.1	n/a	7	contralateral	2.80
	54.4	right	8	ipsilateral	6.70
	54.4	n/a	8	contralateral	1.20
	79.3	right	6	ipsilateral	11.40
	79.3	n/a	6	contralateral	n/a
Dorsal Hippocampus, dHPC	13.2	right	6	ipsilateral	8.10
	13.2	left	6	ipsilateral	1.73 *
	20.1	right	4	ipsilateral	6.80
	20.1	left	4	ipsilateral	0.91 *
	36.1	right	7	ipsilateral	1.6 *
	36.1	n/a	7	contralateral	0.40
	41.1	right	7	ipsilateral	4.00
	41.1	n/a	7	contralateral	0.00
	43.2	right	6	ipsilateral	5.40
	43.2	n/a	6	contralateral	0.50
	83.2	right	6	ipsilateral	8.50
	83.2	n/a	6	contralateral	n/a

TR = transduction time, * ipsilateral injections with <3.5% were assigned as partially infected cases.

## Data Availability

All data are available on request.

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
