# Peer review of "Feasibility of Canine Adenovirus Type 2 (CAV2) Based Vector for the Locus Coeruleus Optogenetic Activation in Non-Transgenic Rats: Implications for Functional Studies"

_brainsci, 2022, doi:10.3390/brainsci12070904_

Round 1

Reviewer 1 Report

Kabanova et al. have performed a rigorous histological assessment of the expression level and tissue damage associated with local and distal targeting of LC-NE neurons using CAV2 expressing ChR-mCherry under the synthetic promoter, PRSx8. This study is a valuable addition to the literature, and carries implications for researchers wishing to study any small nucleus that is characterized as consisting primarily of a single sub-type of cell (e.g. the DA cells in the SN/VTA, or the ACh cells in the NBM) in any non-genetically tractable mammalian species – thus it is of broad interest.

There is one critical omission in the analyses: on line 527 the authors claim “~90% colocalization” (of mCherry in DbH+ cells), and in the intro state that “To validate the cell specificity of the PRSx8 promoter the number of double-labeled (DbH- and mCherry-positive) neurons was normalized to the total number of the mCherry-positive cells.” but the selectivity of the expression is not reported for the local or distal injection groups in the results section. This is an important piece of information for a technical paper such as this.

Minor:

Title – remove ‘the’. Change ‘upregulation’ to ‘expression’ (there are no native opto receptors, and hence nothing to ‘upregulate’).

A more succinct title might be something along the lines of “Canine Adenovirus Type 2 (CAV2) transduction of Locus Coeruleus for optogenetic expression in the rat”

Abstract – final sentence needs to be heavily edited for English/clarity.

Ln 36 – “…over the last several decades.”

Ln 44 – “…enabled the generation of transgenic…”

Ln 53 – tamoxifen and tetracycline are not endogenous ligands! Perhaps the authors mean ‘exogenous’?

Ln 56 – “…several decades,”

Ln 64 – the tamoxifen and tetracycline receptors are also ‘exogenous’, so the word is redundant here. Also, optogenetic receptors can be inhibitory, as well as excitatory.

Ln 65 – “…change the dynamics of brain activity.”.

Ln 73 – 76 If others have used the PRSx8/CAV2 combination in rats successfully previously, what new/useful information is provided in the current manuscript? Was there some short-coming or inconsistency among the previous studies which you have now been able to address?

Ln 82 – “using both local and retrograde transfection strategies.”

Ln 83 – how do you demonstrate that the interaction is ‘competitive’?

Ln 123 – the ‘9’ in ‘10^9’ should be superscript

Ln 136 – ‘have been’ -> ‘were’

Ln 139 – what makes your ‘optothrode’ different from the ‘optrodes’ others have described previously?

Line 159 – ‘has been’ -> ‘was’

Ln 169 – this is usually described as a ‘transcardial’ perfusion, but ‘intracardial’ isn’t wrong.

Ln 187 – ‘glasses’ -> ‘slides’

Ln 221 – ‘whose expression is controlled by’ -> ‘under the control of’

Ln 253 – ‘has been’ -> ‘was’

Ln 347 – why might this be indicative of trans-synaptic transfer? CAV2 is not known to have trans-synaptic properties.

Ln 383 – full stop (period) missing at the end of this line?

Ln 410 – ‘providing’ -> ‘given’. Should ‘NA-containing’ be ‘NE-contaning’?

Ln 421 – ‘accounted’ -> ‘attributed’

Ln 478 – “CAV2 mediated expression requires…”

Ln 477 – 479, AAV expression also requires binding to surface protein receptors and glycans, so this sentence is misleading.

Ln 502 – 503, should this read “can not guarantee…”?

Ln 509 – ‘invasion’ -> ‘transduction’

Ln 517 – the authors claim ~90% colocalization.

Ln 517 – 520, I don’t understand this sentence.

Ln 522 – “our results suggest that detailed”

Ln 525 (and elsewhere) – consider changing ‘neuron elements’ to ‘neurons’.

Ln 530 – 533, mCherry and DbH are visualized using different wavelengths of light, so can you elaborate on how the two interact please?

Ln 544 – “preferential expression of CARs at the axonal terminals makes the CAV2 virus more suitable for a retrograde tracing” but you reported denser staining – a higher proportion of LC-NE neurons transduced – after local injection, so is this statement true?

Ln 553-554 – would it not be more parsimonious to interpret this as uptake from neurons with local projections (and the high local titers may ‘over-ride’ the cell-type selectivity of the promoter)?

Ln 567-569 – yes, perhaps retrograde infection should be considered the preferred method, but with the major caveat that it only labels a relatively sparse selection of cells. Thus, if a researcher’s goal is to silence all NE cells in the LC, they may prefer to accept the compromise of some tissue damage in exchange for greater coverage.  

Fig 1. Panels C and E would be more compelling if the raw data were shown. Without the background activity between APs the reader cannot gauge the SNR of the recordings.

Fig 2. ‘Regardless of the transfection time, there were cases with high (> 70%) and intermediate (~ 50%) infection rates’ – however the fig doesn’t appear to show any intermediate infections at 1 week?

Table 1. Rats 26.2 and 112.2 have injections reported in vLC, but distance to LC is reported as cLC – this is confusing. Also 21.2 was injected in mLC which is reported as ‘cLC’ from the LC, 18.1 was injected in vLC which is ‘<150 um’ from the LC, and 1103 was injected in dLC, which is cLC from the LC. Please clarify.

Table 2. If 21.1 received bilateral injections, how can ‘contralateral’ expression be measured?

Author Response

Reviewer 1. Comments and Suggestions for Authors

Kabanova et al. have performed a rigorous histological assessment of the expression level and tissue damage associated with local and distal targeting of LC-NE neurons using CAV2 expressing ChR-mCherry under the synthetic promoter, PRSx8. This study is a valuable addition to the literature, and carries implications for researchers wishing to study any small nucleus that is characterized as consisting primarily of a single sub-type of cell (e.g. the DA cells in the SN/VTA, or the ACh cells in the NBM) in any non-genetically tractable mammalian species – thus it is of broad interest.

  • We thank the reviewer for the positive feedback and appreciation of our present study. We are also grateful to the reviewer for thorough reading and helpful suggestions for the language corrections. We made an additional effort to improve the language by consulting with a native speaker.

There is one critical omission in the analyses: on line 527 the authors claim “~90% colocalization” (of mCherry in DbH+ cells), and in the intro state that “To validate the cell specificity of the PRSx8 promoter the number of double-labeled (DbH- and mCherry-positive) neurons was normalized to the total number of the mCherry-positive cells.” but the selectivity of the expression is not reported for the local or distal injection groups in the results section. This is an important piece of information for a technical paper such as this.

  • We thank the reviewer for pointing out the missing information about the promoter selectivity quantification. We added the following sentence in the Results (Lines 319-321): “…. Despite substantial between-subject variability in the proportion of virus transduced LC-NE neurons compared to the total number of the DbH-positive cells in the intact LC, the expression of the synthetic PRSx8 promoter, was highly selective and largely restricted to the NE neurons (87.4 ± 4.7% of all mCherry-positive cells within LC) …

Minor:

Title – remove ‘the’. Change ‘upregulation’ to ‘expression’ (there are no native opto receptors, and hence nothing to ‘upregulate’). A more succinct title might be something along the lines of “Canine Adenovirus Type 2 (CAV2) transduction of Locus Coeruleus for optogenetic expression in the rat”

In the title, we referred to the manipulation leading to ‘upregulation’ of the LC activity and not the receptor expression. We revised the title to avoid possible misinterpretation as follows: Feasibility of Canine Adenovirus Type 2 (CAV2) based vector for the Locus Coeruleus optogenetic activation in non-transgenic rats: implications for functional studies

Abstract – final sentence needs to be heavily edited for English/clarity.

  • We thank the Reviewer for this notion. We revised the last sentence and split it into two sentences for clarity. Now it reads as follows: “…The actual fraction of manipulated neurons appears most essential for an adequate interpretation of the study outcome. Thus, besides the cell-type specificity and the transduction efficiency, the between-subject variability in the proportion of remaining viral-transduced targeted cell population must be considered in any functional connectivity study.”

Ln 36 – “…over the last several decades.”

  • We revised it to read “… research over the last several decades…”

Ln 44 – “…enabled the generation of transgenic…”

  • We revised it to read “…enabled the generation of transgenic…”

Ln 53 – tamoxifen and tetracycline are not endogenous ligands! Perhaps the authors mean ‘exogenous’?

  • Corrected to exogenous

Ln 56 – “…several decades,”

  • Revised to read “… Over the last several decades…”

Ln 64 – the tamoxifen and tetracycline receptors are also ‘exogenous’, so the word is redundant here. Also, optogenetic receptors can be inhibitory, as well as excitatory.

We thank the reviewer for pointing out a typo. We changed ‘endogenous’ to ‘exogenous’ ligand. We understand and agree with the reviewer that the light-sensitive receptors can be both, excitatory and inhibitory. However, the focus of this study was on the excitatory type (ChR2), we revise the text to include bidirectional manipulations.  In the Introduction (Lines 69-70), we suggest using optogenetics as an alternative approach to the loss or gain of function experiments in non-transgenic animals: “… The optogenetic tools present an alternative approach to the loss or gain of function experiments in non-transgenic animals. This method employs the use of exogenous light-sensitive receptors to transiently and bidirectionally modulate selective neural circuits…”   

Ln 65 – “…change the dynamics of brain activity.”

  • We revised this sentence to read as follows: “…This method employs the use of exogenous light-sensitive receptors to transiently and bidirectionally modulate selective neural circuits…”

Ln 73 – 76 If others have used the PRSx8/CAV2 combination in rats successfully previously, what new/useful information is provided in the current manuscript? Was there some short-coming or inconsistency among the previous studies which you have now been able to address?

  • The PRSx8/CAV2 combination has been suggested to use for transducing NE neurons for a couple of decades (Hwang, Carlezon et al. 2001, Hwang, Hwang et al. 2005). To date, the CAV2-PRSx8 vector has been successfully applied in non-transgenic rats (Li, Hickey et al. 2016, Swift, Gross et al. 2018, Xiang, Harel et al. 2019, Hayat, Regev et al. 2020). To the best of our knowledge, existing studies using CAV2-based vectors for manipulating LC-NE neurons have not performed systematic quantitative histological analysis. Despite some advantages of using the CAV2xPRSx8 combination (see Discussion, lines 501-514), the AAV-based vectors remain mostly used, while the studies employing PRSx8/CAV2 are rather rare. We believe that quantitative histology shall be a prerequisite for any study employing opto- or chemo-genetic tools. Our present study clearly illustrates that the between-subject variability in the proportion of transduced cells compared to the total size of the targeted population is an important factor (beyond the transduction efficiency and promotor specificity) and shall be considered, particularly for functional studies.

Ln 82 – “using both local and retrograde transfection strategies.”

  • Corrected as “…using both local and retrograde transduction strategy...”

Ln 83 – how do you demonstrate that the interaction is ‘competitive’?

  • We agree with the reviewer. Our data are only suggestive of competitive interactions between ChR2 and DbH expression (Hwang et al., 2005), but do not demonstrate it directly. Therefore, we removed this statement from the Introduction.

Ln 123 – the ‘9’ in ‘10^9’ should be superscript

  • corrected

Ln 136 – ‘have been’ -> ‘were’

  • corrected

Ln 139 – what makes your ‘optothrode’ different from the ‘optrodes’ others have described previously?

  • We apologize for the misspelling. We used a self-made optrode that is similar in principle to others. The distance between the recording site and the end of the optic fiber was custom adjusted to ~1mm.

Line 159 – ‘has been’ -> ‘was’

  • corrected

Ln 169 – this is usually described as a ‘transcardial’ perfusion, but ‘intracardial’ isn’t wrong.

  • corrected

Ln 187 – ‘glasses’ -> ‘slides’

  • corrected

Ln 221 – ‘whose expression is controlled by’ -> ‘under the control of’

  • corrected

Ln 253 – ‘has been’ -> ‘was’

  • corrected

Ln 347 – why might this be indicative of trans-synaptic transfer? CAV2 is not known to have trans-synaptic properties.

We thank the reviewer for this notion. We agree that we did not provide sufficient data to confirm the trans-synaptic transfer of the CAV2 virus and our data suggest a rather non-specific virus expression in non-NE neurons. We corrected the text as follows:” … Such labeling could indicate the virus capacity for the non-specific labeling of non-NE neurons.”

Ln 383 – full stop (period) missing at the end of this line?

  • corrected

Ln 410 – ‘providing’ -> ‘given’. Should ‘NA-containing’ be ‘NE-contaning’?

  • corrected

Ln 421 – ‘accounted’ -> ‘attributed’

  • corrected

Ln 478 – “CAV2 mediated expression requires…”

  • corrected

Ln 477 – 479, AAV expression also requires binding to surface protein receptors and glycans, so this sentence is misleading.

  • We thank the reviewer for this notion. In the Discussion, we focus on the apparent differences between AAVs and CAV-2s. We revised this sentence to make it more clear: “… Whereas AAVs transduce neuron soma to express gene transcripts, CARs are predominantly located on the axon surface [29] making CAV2-based vectors more suitable for the retrograde transduction of selected neuron populations both in the forebrain and spinal cord [29,54,55].”

Ln 502 – 503, should this read “can not guarantee…”?

  • corrected

Ln 509 – ‘invasion’ -> ‘transduction’

  • corrected

Ln 517 – the authors claim ~90% colocalization.

  • We thank the reviewer for pointing out the missing information. We added the sentence about the quantification of the promotor specificity in the Results part (Lines 319-321). The revised sentence reads as follows: “…. Despite substantial between-subject variability in the proportion of virus transduced LC-NE neurons compared to the total number of the DbH-positive cells in the intact LC, the expression of the synthetic PRSx8 promoter, was highly selective and largely restricted to the NE neurons (87.4 ± 4.7% of all mCherry-positive cells within LC).”

Ln 517 – 520, I don’t understand this sentence.

  • We revised this sentence by splitting it into two sentences for better clarity. Now it reads as follows: “…The most important result of this study was the demonstration that the high cell specificity of viral transduction is not sufficient to evaluate the cell population modulated by an exogenous ligand. Our results showed that the number of surviving opsin-expressing neurons can vary significantly between subjects, as well as from the population size in the intact brain.”

Ln 522 – “our results suggest that detailed”

  • corrected

Ln 525 (and elsewhere) – consider changing ‘neuron elements’ to ‘neurons’.

  • corrected

Ln 530 – 533, mCherry and DbH are visualized using different wavelengths of light, so can you elaborate on how the two interact please?

  • We believe there is a misunderstanding. We compared the DbH expression in the mCherry-expressing and intact LC neurons. Consistently reduced fluorescent intensity was suggestive of DbH downregulation due to the virus.

Ln 544 – “preferential expression of CARs at the axonal terminals makes the CAV2 virus more suitable for a retrograde tracing” but you reported denser staining – a higher proportion of LC-NE neurons transduced – after local injection, so is this statement true?

  • In our present study, we did not compare directly the somatic and axonal transduction of LC-NE neurons by CAV2. In the discussion (lines 501-514), we mention some potential advantages of using CAVs vs AAVs. The CAV2 virus was originally proposed as a retrograde tracer because of the expression of the CAR along the surface of axons (Bru, Salinas et al. 2010). Furthermore, Zussy and colleagues showed that CAV-2 can efficiently enter a neuron via presynaptic terminals, as well as other entry sites (Zussy, Loustalot et al. 2016). We believe that the above-mentioned features of the CAV2 do not contradict our findings.

Ln 553-554 – would it not be more parsimonious to interpret this as uptake from neurons with local projections (and the high local titers may ‘over-ride’ the cell-type selectivity of the promoter)?

  • We thank the reviewer for this suggestion. We agree that the uptake from locally projecting neurons could explain our observation. However, we are not aware of NE-containing neurons in the cortex, while the Phox2-expressing cortical neurons have been shown in mice (Allen Brain Institute). In our view, sparse labeling on non-NE cells may indicate a somewhat limited promoter specificity due to the Phox2 binding sites outside the LC area.  We added the following sentence: “…Thus, sparse labeling of non-NE cells in the cortex may indicate a somewhat limited promoter specificity due to the Phox2 binding sites outside the LC area.”

Ln 567-569 – yes, perhaps retrograde infection should be considered the preferred method, but with the major caveat that it only labels a relatively sparse selection of cells. Thus, if a researcher’s goal is to silence all NE cells in the LC, they may prefer to accept the compromise of some tissue damage in exchange for greater coverage.  

  • We agree with the reviewer that manipulating the entire LC-NE population presents various challenges, including cell damage. In the latter case, activation/inhibition of the entire nucleus cannot be achieved.

Fig 1. Panels C and E would be more compelling if the raw data were shown. Without the background activity between APs the reader cannot gauge the SNR of the recordings.

  • We revised Figure 1 as suggested.

Fig 2. ‘Regardless of the transfection time, there were cases with high (> 70%) and intermediate (~ 50%) infection rates’ – however the fig doesn’t appear to show any intermediate infections at 1 week?

  • Thank you for pointing it out. Indeed, at 1 week we only had two cases, both with high transduction rates. This result is likely due to the small sample size for this experimental condition. We expect an increase in variability by the transduction rate with a bigger sample size.

Table 1. Rats 26.2 and 112.2 have injections reported in vLC, but distance to LC is reported as cLC – this is confusing. Also 21.2 was injected in mLC which is reported as ‘cLC’ from the LC, 18.1 was injected in vLC which is ‘<150 um’ from the LC, and 1103 was injected in dLC, which is cLC from the LC. Please clarify.

  • We thank the reviewer for bringing this to our attention. We tried our best to simplify Table 1. Indeed, the injection site was in the ventral LC, which is the ventral part of the LC core. Therefore, this injection was considered in the LC core (0um distance). The same applies to other cases. To avoid further misunderstanding, we corrected the indication of the distance to 0um for all cases with the injection within the LC core.

Table 2. If 21.1 received bilateral injections, how can ‘contralateral’ expression be measured?

  • We thank the reviewer for detecting this typo. The injection was unilateral. We have now corrected Table 2.

Reviewer 2 Report

Kabanova and colleagues performed a comprehensive analysis of LC transduction using CAV2 carrying PRSx8 promoter – the study is well designed and the presented data will be interesting for a wide scientific community. Please, see below a few comments:

Methods – Please, provide animal protocol number.

Methods – Please, provide exact coordinates of injections performed in PFC, dHPC and ACC.

Line 305: how was the downregulation of DBH expression quantified? Please, provide the stats and data. Similarly, in Table 1, please, provide a bit more detail how downregulation of DBH expression was quantified.

Line 306: “mCherry expression was present in the cell membrane” – from current images it is impossible to see to where mCherry expression is localized. Please, perform nuclear, cytoplasm or membrane co-stain to show mCherry expression to be indeed localized to the cell membrane and perhaps quantification would be required as well.  

In case of DBH expression reduction – was there an overall increased death of other neuronal and glia cells within LC?

Line 331: “but ~100-150μm outside the LC 331 border” – please, specify where anatomically I in relation to LC (perhaps LC core)

Line 450: “The virus delivery within the LC core or close to the LC cell bodies caused 449 various undesirable outcomes such as cell death” – no data showing cell death were presented.

Author Response

Reviewer 2. Comments and Suggestions for Authors

Kabanova and colleagues performed a comprehensive analysis of LC transduction using CAV2 carrying PRSx8 promoter – the study is well designed and the presented data will be interesting for a wide scientific community. Please, see below a few comments:

-  We thank the reviewer for positive feedback and appreciation of our present study.

Methods – Please, provide animal protocol number.

  • We added the requested information.

Methods – Please, provide exact coordinates of injections performed in PFC, dHPC and ACC.

  • We used the same coordinates for all animals as indicated in the Method section. In rare cases, we had to deviate from the standard coordinates to avoid, for example, a blood vessel. The ambiguous bregma or skull bone junction could be another source for deviation, yet difficult to estimate. “The following coordinates (from Bregma) were used for the ACC (AP: 1.7 - 2.9 mm, ML: 0.4 mm, DV: 1.4 mm), for the PFC (AP: 3.2 - 4.2mm, ML: 0.5-0.8 mm, DV: 3.5 mm), and the HPC (AP: -2.3 - 4.0 mm, ML: 1.9 - 3.4 mm, DV: 2.0 mm).”

Line 305: how was the downregulation of DBH expression quantified? Please, provide the stats and data. Similarly, in Table 1, please, provide a bit more detail how downregulation of DBH expression was quantified.

  • The DbH downregulation was not quantified but was inferred from visual inspection. A reduced fluorescence between the intact and injected sides was evident and consistent across cases. Unfortunately, we cannot provide quantitative data on the fluorescent intensity.

Line 306: “mCherry expression was present in the cell membrane” – from current images it is impossible to see to where mCherry expression is localized. Please, perform nuclear, cytoplasm or membrane co-stain to show mCherry expression to be indeed localized to the cell membrane and perhaps quantification would be required as well. 

  • We agree with the reviewer and revised this sentence by removing the statement of mCherry expression in the cell membrane.

In case of DBH expression reduction – was there an overall increased death of other neuronal and glia cells within LC?

  • We thank the Reviewer for this important point. Unfortunately, we have not performed glial staining or staining for cell death markers. We agree that quantitative characterization of DbH expression shall be addressed in future studies using the PSRx8 promoter. We added the following sentence in the Discussion: ‘’… Possible virus-induced alternation of the LC-NE neuron phenotype, including apoptosis or DbH expression, shall be addressed in the future studies.”

Line 331: “but ~100-150μm outside the LC 331 border” – please, specify where anatomically I in relation to LC (perhaps LC core)

  • We revised this sentence as suggested: “…~100-150μm outside the LC border and lateral to the mesencephalic trigeminal nucleus (Me5), typically resulted in a good outcome.”

Line 450: “The virus delivery within the LC core or close to the LC cell bodies caused 449 various undesirable outcomes such as cell death” – no data showing cell death were presented.

- We added images showing severe LC damage and DbH downregulation to Fig 2.

References

Bru, T., S. Salinas and E. J. Kremer (2010). "An update on canine adenovirus type 2 and its vectors." Viruses 2(9): 2134-2153.

Hayat, H., N. Regev, N. Matosevich, A. Sales, E. Paredes-Rodriguez, A. J. Krom, L. Bergman, Y. Li, M. Lavigne, E. J. Kremer, O. Yizhar, A. E. Pickering and Y. Nir (2020). "Locus coeruleus norepinephrine activity mediates sensory-evoked awakenings from sleep." Sci Adv 6(15): eaaz4232.

Hwang, D. Y., W. A. Carlezon, O. Isacson and K. S. Kim (2001). "A high-efficiency synthetic expression selectively promoter that drives transgene in noradrenergic neurons." Human Gene Therapy 12(14): 1731-1740.

Hwang, D. Y., M. M. Hwang, H. S. Kim and K. S. Kim (2005). "Genetically engineered dopamine beta-hydroxylase gene promoters with better PHOX2-binding sites drive significantly enhanced transgene expression in a noradrenergic cell-specific manner." Mol Ther 11(1): 132-141.

Li, Y., L. Hickey, R. Perrins, E. Werlen, A. A. Patel, S. Hirschberg, M. W. Jones, S. Salinas, E. J. Kremer and A. E. Pickering (2016). "Retrograde optogenetic characterization of the pontospinal module of the locus coeruleus with a canine adenoviral vector." Brain Research 1641: 274-290.

Swift, K. M., B. A. Gross, M. A. Frazer, D. S. Bauer, K. J. D. Clark, E. M. Vazey, G. Aston-Jones, Y. Li, A. E. Pickering, S. J. Sara and G. R. Poe (2018). "Abnormal Locus Coeruleus Sleep Activity Alters Sleep Signatures of Memory Consolidation and Impairs Place Cell Stability and Spatial Memory." Curr Biol 28(22): 3599-3609 e3594.

Xiang, L., A. Harel, H. Gao, A. E. Pickering, S. J. Sara and S. I. Wiener (2019). "Behavioral correlates of activity of optogenetically identified locus coeruleus noradrenergic neurons in rats performing T-maze tasks." Scientific Reports 9(1): 1361.

Zussy, C., F. Loustalot, F. Junyent, F. Gardoni, C. Bories, J. Valero, M. G. Desarmenien, F. Bernex, D. Henaff, N. Bayo-Puxan, J. W. Chen, N. Lonjon, Y. de Koninck, J. O. Malva, J. M. Bergelson, M. di Luca, G. Schiavo, S. Salinas and E. J. Kremer (2016). "Coxsackievirus Adenovirus Receptor Loss Impairs Adult Neurogenesis, Synapse Content, and Hippocampus Plasticity." J Neurosci 36(37): 9558-9571.

Round 2

Reviewer 1 Report

The authors have completed extensive revisions that have addressed all of my concerns.